# A Novel Modality Enables New Evidence-Based Individual Risk Stratification That Can Potentially Lead to Decisive Management and Treatment Decisions in Prostate Cancer

**DOI:** 10.3390/diagnostics13030424

**Published:** 2023-01-24

**Authors:** Meir Weksler, Avi Simon, Robert E. Lenkinski, Hagar Landsman, Haim Matzkin, Nicola Mabjeesh, Ilan Leibovitch

**Affiliations:** 1R&D Department, Prosight Ltd., Bay-Yam 5697439, Israel; 2Department of Radiology, University of Texas Southwestern Medical Center, Dallas, TX 75390, USA; 3Physics Department, Weizmann Institute of Science, Rehovot 7610001, Israel; 4Department of Urology, Tel Aviv Sourasky Medical Center, Sackler Faculty of Medicine, Tel Aviv University, Tel Aviv 69978, Israel; 5Urology Department, Soroka Medical Center, Beer Sheva 84101, Israel; 6Urology Department, Meir Medical Center, Kfar Sava 4428164, Israel

**Keywords:** cell proliferation, epithelial glands, interactive biopsy guidance, malignancy score, risk stratification, individual treatment recommendations

## Abstract

A key step in providing management/treatment options to men with suspected prostate cancer (PCa) is categorizing the risk in terms of the presence of benign, low-risk, intermediate-risk, or high-risk disease. Our novel modality brings new evidence, based on the long-known hallmark characteristic of PCa—decreased zinc (Zn), which is the most direct metabolic sign of malignancy and its aggressiveness. To date, this approach has not been adopted for clinical use for a number of reasons that are described in this article, and which have been addressed by our approach. Zn has to be measured on fresh samples, prior to fixating in formalin; therefore, samples have to be scanned during the biopsy session. As Zn depletion occurs in the glands where the tumors develop, estimation of the glands’ levels in the scanned tissue, along with their compactness, are essential for accurate diagnosis. Combined with the Zn depletion, this facilitates a reliable assessment of disease aggressiveness. Data gathered in the clinical study described here indicate that, in addition to improving the biopsy quality by real-time interactive guidance, a malignancy score can now be established for the entire prostate, allowing higher granularity personalized risk stratification and more decisive treatment decisions for all PCa patients.

## 1. Introduction

### 1.1. Prostate Cancer Background 

Prostate cancer (PCa) is the most common cancer among non-smoking men with an incidence rate of 60% in men over 65. Early detection and accurate risk stratification are essential for providing appropriate management/treatment options leading to reduced morbidity and mortality. Delayed and/or inappropriate treatment can lead to the evolution of more advanced disease, such as metastases to the bones or bladder and, eventually, death. 

Current diagnosis consists of screening, involving PSA blood tests, followed by needle biopsy, where 12–14 cores are typically extracted and sent for histopathology, which has the largest weight in treatment decision making today. 

About two million biopsies per year are performed in the US and Europe combined [1]. 

### 1.2. Zinc Depletion as a PCa Marker 

PCa stakeholders agree that “New biomarkers are needed to avoid unnecessary biopsies and radical prostatectomies to distinguish benign from malignant lesions and to better discriminate localized from advanced disease.” [2] (p. 1355). A known biomarker—depletion of Zn in the epithelial glands of the prostate—has already been used in research for a few decades but has not yet been applied to clinical practice. 

“It has been clinically established that markedly decreased zinc concentration in prostate cancer tissue and prostatic fluid, compared to normal and benign prostate, exists in virtually all cases of PCa, and, thereby, constitutes the most consistent hallmark signature identification of PCa. The decrease in zinc is an early event in the development of malignancy and persists in progressing malignancy. These relationships offer the opportunity for an accurate specific biomarker.” [3] (pp. 11,12). In spite of this powerful citation and other research [4,5,6,7], these important findings are within the domain of researchers and, until now, have not been incorporated into clinical routines for PCa diagnosis. 

### 1.3. X-ray Fluorescence (XRF) 

Traditional qualitative microscopy methods of Zn detection are far from being sufficiently accurate for PCa grading, nor can they avoid impacting the clinical workflow. 

XRF scanning is used for very accurate non-contact measurement of trace elements. It is in use also for several medical applications, such as in vivo detection of lead (Pb) in bone, cadmium (Cd) in the kidney, mercury (Hg) in the brain, and iodine (I) in the thyroid. XRF is a proven non-destructive and safe technology, most appropriate for detecting small quantities of elements, including Zn. Since biopsy samples sent to histopathology are fixated in formalin and lose the Zn information, XRF scanning must be performed on fresh biopsy samples during the biopsy session to achieve accurate quantitative measurements, without compromising the workflow or causing delays in the biopsy session. 

Scientists at the Weizmann Institute of Science in Rehovot, Israel (WIS) have conducted an extensive study at Sheba Medical Center in Ramat Gan, Israel [8,9], using XRF for scanning fresh biopsy samples from over 600 patients (over 2400 biopsy cores) in order to establish, for the first time, that zinc depletion is proportionate to the severity (grading) of PCa. The results have consolidated the correlation between PCa aggressiveness and zinc depletion. However, they did not have the required accuracy and resolution needed for assessing individual PCa aggressiveness. 

## 2. Materials and Methods

This article elaborates on a novel modality, ScoRisk, for accurate prostate cancer (PCa) aggressiveness scoring, risk stratification and treatment decision support, based on Zn depletion in the PCa epithelial glands and related proliferation changes in tissue parameters (covered in one granted patent and two pending patents).

### 2.1. X-ray Fluorescence (XRF) for ScoRisk 

XRF scanning provided the data described in the graph of Figure 1 where: Zn = zinc count; C = Compton count (inelastic scattering), which is mainly indicative of tissue quantity; and C_0_ = C + (the elastic scattering component), which includes the component of the K-lines of the rhodium target in the X-ray tube and is indicative of cell proliferation/density/compactness. The difference ΔC= C_0_ − C is a direct surrogate measure of the amount of epithelial glands in the tissue.

As there is considerable non-uniformity in the thickness of the biopsy samples while scanning, C_0_ is used for normalizing the Zn count, as, due to the elastic scattering component, it includes a more representative indication of overall tissue quantity/proliferation/density than C. To facilitate this, we use a 45^0^/45^0^ source/detector configuration. 

C_0_ is also crucial for deriving personal tissue information, indicative of molecular structure, cell proliferation and density/compactness, changes in which occur in the epithelial glands when malignancy starts and which increase as tumors develop [10]. 

### 2.2. The Souraski Clinical Study 

The purpose of the study was to gather XRF tissue and Zn data from fresh biopsy samples on parameters indicative of PCa for the development of a ScoRisk algorithm. 

#### 2.2.1. Equipment

An off-the-shelf XRF spectrometer (M4 Tornado from Bruker GmbH, Berlin, Germany), modified and adapted to our specific needs, was utilized. The modifications included beam size, intensity and scan time that allowed, in due course, routine use of the final product during the biopsy session. For the purpose of this study, real-time was not essential.

The beam diameter was 2 mm, which at 45^0^ resulted in a spot size of 2 mm by 3 mm in the scan direction (the 2 mm diameter covered for the typically ~1 mm wide sample that is often distorted while releasing the sample from the biopsy needle). The pitch (pixel) was 1 mm. A dedicated low-background plastic tray, with mylar openings was used for placing the samples at predefined, numbered intervals. An X-Y-Z stage was installed, including a dedicated tray holder. It permitted moving the tray holder to the loading station and then realigning and focusing the samples on the tray under the beam prior to start of the scan (Figure 2).

Once the sample was extracted from the patient, the nurse used tweezers to take it out from the biopsy needle and locate it on the tray along a marked line and between two marked segments along the line. Then the tray was located on the XYZ stage that was set up in the loading/unloading position. On pressing start-of-scan, the XRF was sealed and the XYZ stage took the tray under the beam to start scanning according to a preprogrammed scheme. Important note: In the final product, removal of the sample from the biopsy needle and locating it on the tray will be performed in a dedicated third-party automatic device (already FDA-approved). For further details regarding XRF, see reference [11]. 

#### 2.2.2. Study Routine

Upon extraction, the biopsy samples were placed by the urologist’s assistant in numbered grooves onto the plastic tray. The tray with several samples was then placed onto the XRF tray holder; all samples were scanned in batches and subsequently placed in individual formalin bottles, as routinely performed during biopsy, and then sent to histopathology for diagnosis. 

#### 2.2.3. Pathology Report

The pathology report was at the level of the sample. The diagnosis for each sample was provided according to the Gleason grading scheme. For the sample level, histopathology diagnosis was provided according to the then used Gleason grading and scoring scheme for PCa. The newer sample grading scheme by Epstein (GG1 to GG5) was not used here in order to avoid confusion with Epstein’s prostate grading scheme (G1 to G5, used by Prosight in this article for scoring the prostate malignancy). 

For negative (benign) samples, pathology provided the type of tissue (SGH, FMH) and/or metabolic condition (BCH, TCM, AA, MCI and prostatitis). Inflammation, fibro-muscular hypertrophy (FMH), stroma and glandular hyperplasia (SGH), FMH or SGH + metabolic (atrophic acini, basal cell hyperplasia, transition cell metaplasia, mild chronic inflammation) were indicated.

For positive (PCa) samples, pathology provided the sample score. For example: g6(3 + 3) score, corresponding to two g3 tumor grades in the sample; g7(4 + 3), corresponding to g4 and g3 tumor grades in the sample, etc. 

Positive sample scores encountered in the study included: g6(3 + 3), g7(g3 + 4); g7(g4 + 3), and g8(g4 + 4). Since no g5 tumor grade was encountered, g9 and g10 samples scores are not listed. The cancerous portion in the samples varied from 5% to 90%. A total of 320 biopsy samples (cores) from 20 patients with 16 biopsy cores each (about 3200 pixels) were scanned in batches as described above. 

### 2.3. Data Analysis and Results 

#### 2.3.1. Background

In prostates, Zn accumulates mainly in the epithelial gland cells. Each sample/pixel contains variable amounts of epithelial gland cells within the stroma and/or fibromuscular supporting tissue. The amount of Zn in benign prostate tissue is typically linearly dependent on the amount of gland cells in that tissue. The linear relationship between tissue and Zn values for normal tissue (i.e., non-cancerous or other metabolic tissue) indicates that the tissue value in each pixel incorporates the relative amount of epithelial gland cells in that pixel. 

PCa develops predominantly in epithelial glands when the Zn is depleted and tissue proliferation is induced. Consequently, the Zn/C_0_ ratio for PCa pixels/samples will be smaller than that for benign pixels/samples.

#### 2.3.2. Analysis Observations

During analysis of the data, the following was observed:Zn levels vary substantially among prostates and, therefore, a population (all prostates, all cores, and all pixels) normalization scheme was introduced.Each prostate has a unique malignancy signature/score (akin to Epstein’s prostate grade [12]), which is dependent not only on the Zn content, but also on tissue characteristics typical of PCa—oncogenic manifestations of neoplasia/hyperplasia in glands and tissue proliferation mechanisms, which are known indicators of the malignancy processes.

It is important to note the distinction between a low level of Zn and an actual cancer-typical Zn depletion in a biopsy sample or pixel. 

#### 2.3.3. Analysis Steps—Scanning and Data Generation 

During scanning, Zn, C and C_0_ values for each pixel were acquired. As discussed above, C is used for volume and density normalization of Zn, and C_0_, which includes C, is used as a measure of tissue proliferation.

The following parameters were then generated for each pixel, sample and prostate: ΔC= C_0_ − C, which is a measure of the relative amount of epithelial glands in the tissue; Zn/C; and Zn/C_0_, which is the amount of Zn normalized to tissue proliferation.

A population database, comprised of the statistical population levels for each of the parameters was then generated from the data from all the pixels from all the samples, from all the individual prostates scanned by the system, following data cleaning (see below).

Subsequently, the values for each pixel, sample and individual prostate were normalized to the respective population levels. 

#### 2.3.4. Data Cleaning—Population and Prostate Levels

The normalized population level data were cleaned to remove invalid pixels: non-uniform, or certain edge pixels, or pixels that were disconnected from the sample, or very small and/or very thin pixels, typically characterized by very small C and very small Zn/C.Pixels with very small ΔC, indicative of low or no glandular tissue.

For each individual prostate, standard statistical steps were used to remove outliers at the level of the prostate: pixels with very small C and/or ΔC and/or Zn/C.

#### 2.3.5. Prostates—Population Data

When establishing the PCa aggressiveness of the prostate, the malignancy score, the parameters for each prostate were normalized and compared to the corresponding population parameters. Table 1 below presents the current average values of the population parameters database (not including two prostates with prostatitis that we detected as such). 

### 2.4. Prostates Aggressiveness—Malignancy Score

In the standard-of-care prostate grading scheme by Epstein [12], the lowest level is G1 and the highest is G5. Our prostate malignancy scoring scheme differentiates between G0 (not available in Epstein’s prostate grading scheme) and G1 in two aspects: G0 corresponds to prostates where no positive samples were identified during the biopsy due to either the absence of tissue proliferation or Zn depletion or both at the prostate and core levels; while,G1 corresponds to a prostate where tissue proliferation (C_0_) and Zn depletion for the prostate were around the population average and only a few samples had levels of C_0_ and Zn depletion (Zn/C_0_) characteristic of low-malignancy-score PCa.

This prostate aggressiveness scoring distinction is important in relation to early detection and the type of active surveillance that the urologist recommends for such patients.

#### 2.4.1. Prostate Malignancy Score—Diagnosis Results

Prostate characterization during a biopsy is possible since Zn, C_0_ and Zn/C_0_ for negative prostates are significantly different than the same characteristics in positive prostates. This is reflected in Figure 3 below.

There is a strong and evident differentiation between the XRF levels of C_0_, and Zn/C_0_ characteristics for high-PCa-malignancy-score prostates and those for benign or low-malignancy-score prostates. The levels of these parameters are also dependent on whether the Zn level in the prostate is low, mid or high. It was found that:For low-/mid-Zn prostates (positive):
●The mean C_0_ for PCa samples classes was significantly higher than those for the benign classes.●The mean Zn/C_0_ for PCa samples classes was smaller (and, for the majority of the classes, much smaller) than for the benign sample classes.
For high-Zn prostates:●High-Zn PCa prostates had much higher C_0_ (tissue proliferation) values than those for high-Zn benign prostates.


Additionally: ●High malignancy score in a prostate was also correlated with a high number of positive cores in the prostate. ●Decrease in Zn/C_0_, in tandem with high tissue proliferation (C_0_), was indicative of a higher malignancy score.

#### 2.4.2. Pixels Grading 

Once the analysis for the prostate was generated, all population-normalized pixel parameters were classified in relation to the prostate parameters. Pixels with very low C, very low ΔC and very low Zn/C_0_ relative to the respective prostate levels, were cleaned out of the sample analysis. The following are some global rules for determining the pixel level grade:High Zn/C_0_ → no Zn depletion → non-PCa pixel.Very low C_0_ → no tissue proliferation → non-PCa pixel.Mid or low Zn/C_0_ and mid/high C_0_ → PCa pixel.

Below in Figure 4 and Figure 5 are two examples from the scanned prostates.

The positive pixels—g3, in this case—had a C_0_ (x-axis) threshold level = 1.07 (for this specific prostate), and had values of C_0_ and Zn/C_0_ (y-axis) bound between the two Zn threshold lines for this specific prostate, with a lower threshold between g3 and outlier low pixels corresponding to a Zn value of 0.63, and an upper threshold between g3 and benign pixels corresponding to a Zn value of 1.07.

The same logic applies to the high-PCa-malignancy-score (G4) prostate below. 

The positive pixels—g3 and g4, in this case—had a common C_0_ threshold level = 0.97 (for this specific prostate), and had values of C_0_ and Zn/C_0_ bounded between the three Zn threshold lines: a lower threshold boundary between g4 and outlier low pixels corresponding to a Zn value of 0.45; a middle threshold boundary between g4 and g3 pixels corresponding to a Zn value of 0.66; and an upper threshold boundary between g3 and benign pixels corresponding to a Zn value of 0.91. 

Regarding the “pre-PCa” pixels in both graphs above: since Zn depletion precedes the morphology changes, it seems reasonable that in the near future, the algorithm will have the ability to identify such cases and further improve sensitivity.

Figure 4 and Figure 5, therefore, demonstrate our ability to separate individual grades, since Gleason g3 and g4 pixels are clearly identified and classified. Accordingly, samples containing only g3 lesions would correspond to the new Gleason Grade Group1—GG1 (3 + 3), while samples containing g3 and g4 lesions would correspond to either GG2 (3 + 4) or GG3 (4 + 3), and samples containing only g4 lesions would correspond to GG4 (4 + 4).

Such a “template” of thresholds is generated for each prostate. 

#### 2.4.3. Biopsy Metrics 

The accuracy of our non-blinded analysis was measured vis-a-vis the pathology findings that served as an absolute reference against which the analysis quality metrics were calculated. Below, in Table 2, all the biopsy quality metrics are tabulated.

As the Zn depletion precedes the morphology changes on which the pathology findings are based, note that, at the sample level, FP/P is much lower than FN/N (about 10% vs. 3%).

## 3. Considerations on the Routine Use of ScoRisk in the Clinic

Unlike the clinical study, where real-time grading and guidance was not required and samples were scanned in batches, during routine use in the clinic, the biopsy samples are introduced one-by-one into a desktop X-ray fluorescence (XRF) device for scanning, immediately after removal from the prostate. 

While the urologist removes the next sample, the grades/score of the previous sample is graphically displayed on the ultra-sound (US) screen (which is used in ALL types of biopsies), superimposed on the US image, in tandem with the “rolling/evolving” prostate malignancy score, both along with the statistical certainty; the more samples that are extracted, the higher the certainty. 

After scanning, the samples are placed in formalin bottles for histopathology as routinely performed, with negligible impact on the flow and without losing time.

### 3.1. Flowchart—as Part of Routine Clinical Flow

The flowchart in Figure 6 below outlines the real-time algorithm of ScoRisk that is to be used during a biopsy for prostate malignancy scoring, sample/pixel grading and during biopsy interactive guidance. 

Additional details are described below the flowchart, according to the numbering of selected blocks. 

**Block 0:** Database. The following is included in the database for reference during the biopsy session: data (block 1), parameters (block 2); pathological conditions, as gathered from selected sites, following QC; pixels and sample grades and prostate malignancy scores; compartmentation (such as ethnicity, age, etc.); clinical data that might improve the accuracy in due course (such as PSA, PSA density, PSA velocity, etc.). 

**Block 1:** Scan + Data Generation. The same XRF scanning scheme used in the clinical study is employed following the removal of each sample.

**Block 4:** Compute and Display. When establishing the prostate malignancy score the calculations are at the population level in the DB. When establishing the pixels and sample grades, the classifications are at the combined population and prostate (biopsy) levels.

**Block 5:** Interim Prostate Grading.

Following the removal and scanning of 2 to 3 samples (ca. 10 pixels each) from each side of the prostate (40 to 60 pixels) an interim prostate classification is established according to the malignancy score rules at the prostate level.

Once the analysis for the prostate is generated, all pixel parameters are normalized and classified in relation to the prostate parameters. 

In some clear-cut cases, it could be established quite early in the process that the prostate is benign and that the biopsy could be stopped to avoid unnecessary risk of complications to the patient. 

In cases of MRI-guided biopsy, an additional interim classification could be considered for stopping the biopsy, upon confirming one or two positive g7(4 + 3) samples in an ROI (region of interest).

**Block 9:** Final Prostate Malignancy Score.

The thresholds derived from Table 1, adjusted to the individual prostate, along with the number of positive samples and % of malignant volume, are reiterated and used for final grading of the samples and generating the final malignancy score, from which the individual risk stratification and treatment recommendations are generated in the Final Report (block 10). 

### 3.2. Interactive Biopsy Guidance

Following extraction of 2 to 3 samples from each side, the system starts automatic guidance, suggesting changes in the original plan (which is random in the systematic biopsies). The graph in Figure 7 exemplifies a typical quick convergence of parameters.

Note that, after 2 to 3 samples on the left (L) and right (R) sides, the “rolling” prostate-level averages reach close-to-final values, therefore allowing guidance. It is quite clear that the left side is positive, and the right side is negative. 

Some examples of guidance (not specifically in the case of Figure 7): take one or more samples from a certain area; or move to a different region, if the area is not malignant; or possibly stop the biopsy, in case sufficient reliable information is gathered following guidance and avoid taking additional unnecessary samples. 

Below is an initial partial list of how the urologist could use ScoRisk during-biopsy for a better biopsy outcome and for deriving individual risk stratification and treatment recommendations at the conclusion of the biopsy.

#### 3.2.1. Interactive Guidance Examples

Low prostate malignancy score in tandem with low-grade samples: recommendation to abandon the current region and continue to the next;In case of a mid to high malignancy score in tandem with low- to mid-grade samples: continue taking more samples from the same region to increase the positive yield (cancer volume) and sharpen the index lesion;In the event inflammation (prostatitis) is detected: consider stopping the biopsy and avoid the extraction of unnecessary additional samples.

#### 3.2.2. Personalized Treatment Recommendations

Current NCCN/AUA/ACS guidelines are quite coarse. With ScoRisk, finer and sharper malignancy scores for comparable index lesion samples could lead to different, more-specific personalized treatment recommendations. 

In doubtful cases this offers the possibility to decide more safely that, for example, radical prostatectomy (RP) can be postponed, or in the case of negative biopsy, in support of active surveillance, to postpone or pull-in a repeat biopsy. 

## 4. Discussion

The scientific puzzle of how to take advantage of the best PCa indicator for routine clinical use, Zn depletion, seems to have been solved. All the pieces seem to be in place: a novel measure of prostate aggressiveness, the malignancy score, on top of sample grading; the ability to get and use this information in real-time during the biopsy; the accuracy required for individual aggressiveness determination; and the non-destructive, workflow-neutral nature of the modality.

The analysis clearly corroborates the metabolic signs of PCa, seemingly in advance of any other existing method and with much higher fidelity. The higher accuracy is based on unique characteristics of the new modality: It is a direct measurement of the most reliable prostate cancer malignancy indicator known to science, Zn depletion, accurately measured by XRF.The newly discovered ability of XRF to discriminate tissue-related signs of cancer (including tissue proliferation), which is an additional marker of PCa and further improves the accuracy of ScoRisk.The ability to remove and scan the most suspected samples, thanks to real-time guidance during the biopsy, in order to extract as many positive, or “closest” to positive, biopsy cores.Zn depletion precedes the morphology changes that the current pathology is based on, thereby providing earlier detection.The measurements are on the full depth (3D) of the needle biopsy core, providing inherently more representative results than looking at a 25-micron-thin microscopy slide sample (over 1/50 thinner than the original biopsy core, on the surface of it only).The ability to determine the prostate-level malignancy score, which is a higher granularity prostate-grading capability, especially effective in doubtful cases, where the urologist/oncologist seeks support in decision making, beyond Epstein’s grading scheme. Concurrently, our modality is designed to generate higher granularity sample grading, both through continuous reference to a large, AI-enhanced, continuously updated population database incorporating the various pathologies and the corresponding XRF parameters

The results demonstrate the potential impact of ScoRisk in prostate cancer (PCa) biopsy, risk stratification and treatment management by providing new evidence-based data that will lower uncertainty in many cases. 

Accordingly, in case of doubt, the urologist has additional information at their disposal for getting more decisive and safer decisions at a much lower cost than utilizing expensive and quite inaccurate genomic, or other risk prognosis, methods, that in addition to their inherent inaccuracy, can, in the best case, only be as good as the biopsy sample quality [13,14,15,16,17,18]. Moreover, genomic tests for lower malignancy score prostates are not as meaningful, while ScoRisk can provide accurate information for low-malignancy, and even benign, prostates.

### 4.1. Limitations of the Study

To assess the potential of ScoRisk, we had to start by comparing our results with the histopathology findings. It is clearly not an ideal comparison, as (i) Zn depletion precedes the pathology findings, and (ii) we examine the bulk of the biopsy core, which is more representative than looking at the surface of an ultra-thin microtome. As we developed the algorithms based on the above comparison, well after the clinical study was completed, we could not settle the discrepancies, but we are of the opinion that, in most cases, we would have been more accurate than the histopathology findings. Overcoming these limitations will be among the challenges during the next clinical studies we plan to undertake described below.

### 4.2. Short-Term Plans 

The foundations of a device for clinical use realizing the potential of ScoRisk were laid in this work. The following are non-exhaustive clinical steps and improvements suggested for the device, prior to releasing a commercial product: In addition to the clinical trials for the regulatory steps, conduct on-going systematic retrospective follow-up on treatment recommendations, to ensure the fidelity of the prostate malignancy score in providing improved risk stratification for better treatment recommendations.Implement a real-time guidance and grading digital code, centered on multivariate statistical analysis and deep-learning tools, including a large-scale accessible population database and real-time diagnosis, grading/scoring and guidance during the biopsy.Implement a built-in camera to simultaneously measure the volume for each pixel during the XRF scan to further improve normalization.Take advantage of digital pathology during the next studies/trials, as it is not practical for pathologists to perform pixel-by-pixel analysis.

### 4.3. Long-Term Plans

As Zn depletion precedes the morphological changes, conduct on-going follow-up studies to reinforce ScoRisk’s sample grading as an independent standard by gathering statistical information rather than solely comparing to pathology findings. Introduce a dedicated tool for analyzing a prostate following prostatectomy for completely automatic analysis of whole-mount samples that will be much thicker than today’s microtomes and able to create a much more accurate 3D malignancy map reconstruction of the whole prostate. 

### 4.4. One-Stop-Shop 

We have created the basis for a one-stop-shop clinical routine that translates the most direct PCa severity predictor, Zn depletion, into a single, accurate method for assessing PCa aggressiveness, stratifying individual risk and offering treatment recommendations, to improve any type of biopsy, dramatically reduce repeat biopsies, avoid overtreatment, decrease morbidity and mortality and reduce the overall economic burden of treating PCa. 

## 5. Patents

Granted, US 9,052,319—METHOD OF GUIDING PROSTATE BIOPSY LOCATION—SIMON Avi et al.Pending, WO2021/009762—METHOD AND SYSTEM FOR ANALYZING PROSTATE BIOPSY—WEKSLER Meir et al.Pending, 2022—WEKSLER Meir et al.

## Figures and Tables

**Figure 1 diagnostics-13-00424-f001:**
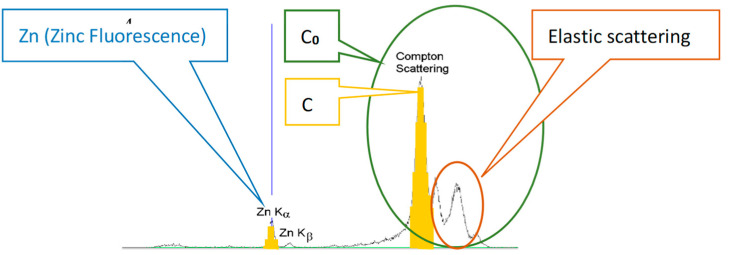
Zn and scattering counts.

**Figure 2 diagnostics-13-00424-f002:**
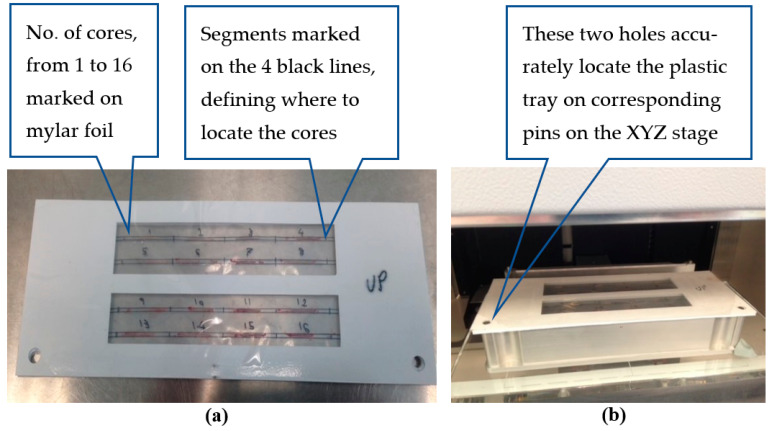
Samples tray and tray holder in the Bruker XRF device. (**a**) Low-background plastic tray; (**b**) Tray holder located on the XYZ stage, with plastic tray on top.

**Figure 3 diagnostics-13-00424-f003:**
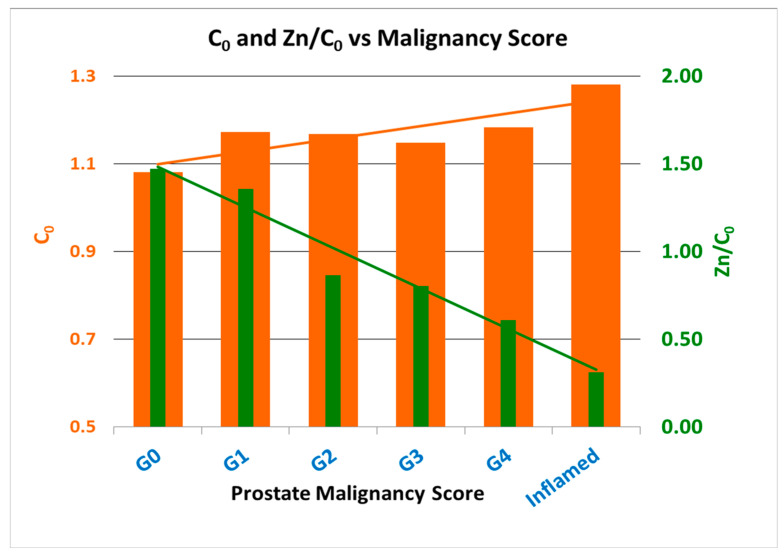
Prostate malignancy scores as a function of the prostate levels of C_0_ (tissue proliferation) and Zn/C_0_ (Zn depletion). No G5 prostates were found in this study.

**Figure 4 diagnostics-13-00424-f004:**
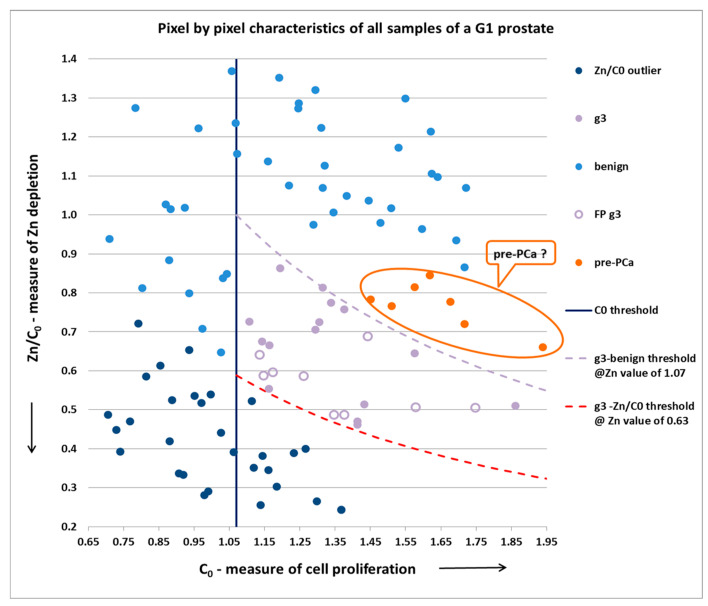
Pixel characteristics and thresholds in a low-PCa-malignancy-score G1 Prostate (9 samples out of 16 were g6(3 + 3), the remainder being only g3 and benign pixels).

**Figure 5 diagnostics-13-00424-f005:**
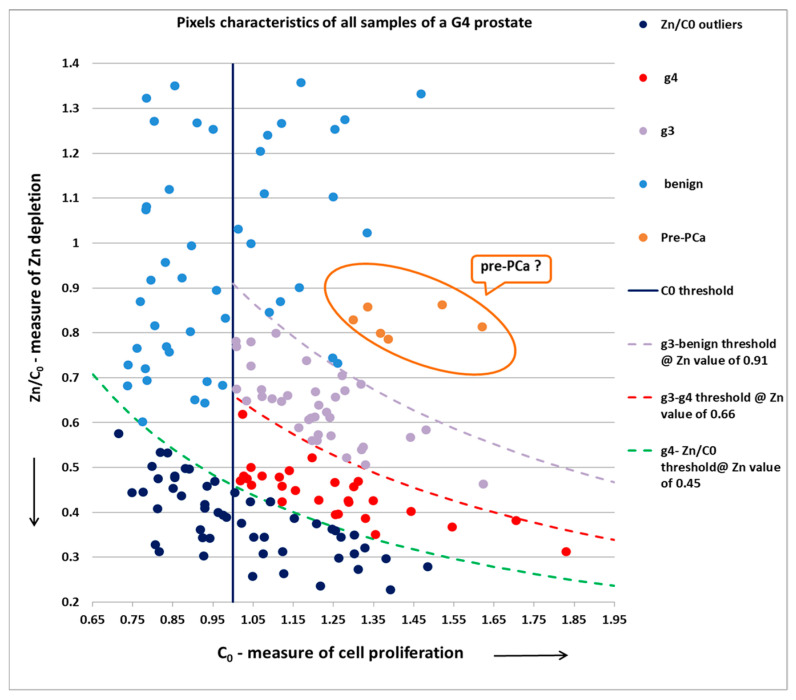
Pixel characteristics and thresholds in a high-PCa-malignancy-score G4 prostate (one single benign sample, and several g7 (3 + 4), g7 (4 + 3) and g8 (4 + 4); pixels: benign, g3 and g4).

**Figure 6 diagnostics-13-00424-f006:**
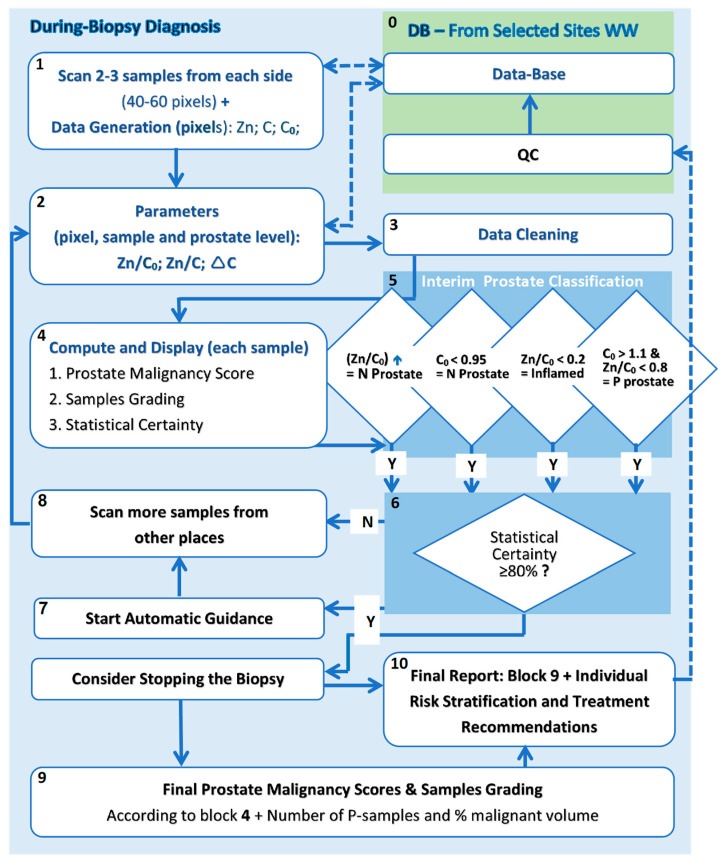
Clinical flowchart—malignancy scoring, sample grading/scoring and guidance during biopsy.

**Figure 7 diagnostics-13-00424-f007:**
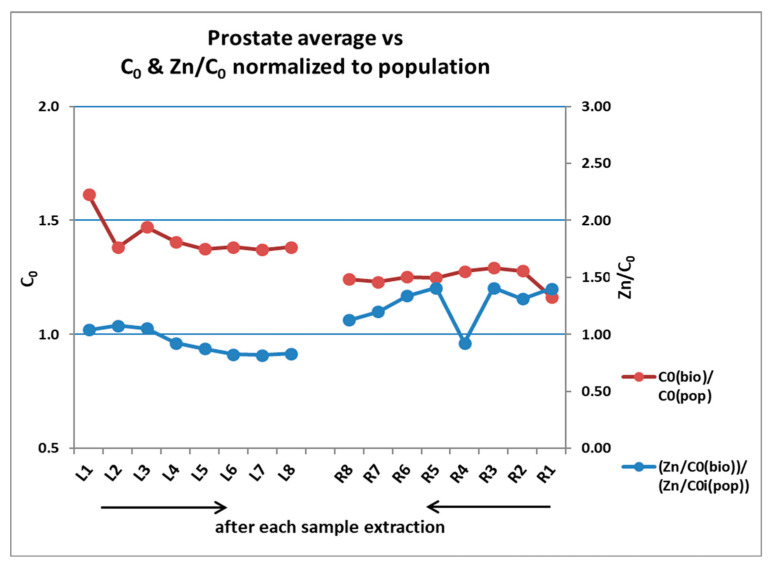
Convergence towards guidance.

**Table 1 diagnostics-13-00424-t001:** Population parameters database.

Statistics	C_0_ (Tissue)	C	ΔC=	Zn	Zn/C	Zn/C_0_
			(C_0_ − C)			
Mean	1.14	1	0.14	1	1.41	1.23
STD	0.24	0.23	0.05	0.76	1.01	0.88
Q1	0.96	0.83	0.1	0.47	0.6	0.6
Median	1.13	0.99	0.13	0.78	1.1	0.97
Q3	1.29	1.15	0.17	1.24	1.76	1.54

Q1—1st quartile; Q3—3rd quartile.

**Table 2 diagnostics-13-00424-t002:** Analysis—Biopsy Quality Metrics.

Metrics	Prostate Level	Sample Level
After data cleaning * P TP (true positive)	20 10 10	308 64 57
N (including Inflamed) TN (true negative)	10 10	244 240
Inflamed prostates (out of TN)	2	32
False N False P Sensitivity	0 0 100%	7 4 89%
Specificity Accuracy	100% 100%	98% 96%

* Samples with very few pixels (<4) and/or missing in the pathology report.

## Data Availability

The raw data, on which this article is based is available upon request.

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
