# Peer review of "A Novel Modality Enables New Evidence-Based Individual Risk Stratification That Can Potentially Lead to Decisive Management and Treatment Decisions in Prostate Cancer"

_diagnostics, 2023, doi:10.3390/diagnostics13030424_

Round 1

Reviewer 1 Report

Materials and Methods:

Line 89-94: need a reference of additional information to understand the derivations of the outlined correlation between XRF and tissue biology.

Line 107: Company name should be deleted and should be corrected as “ScoRisk algorithm”

Line 137: Correct histopathological notations must be used. Authors should identify when they are referring to Gleason grade and Gleason score. Grade can be 1-5 where are score can be 2-10. Correct notation of Gleason score 6 (3+3), 7 (3+4), 7 (4+3) etc. Remove notations g6(33), g7(34), etc.

Results:

Line 213: Figure 3: This shows prostate malignancy score vs C0 vs Zn/C0. How did authors separately calculated C0, Zn/C0 for individual Gleason Grades? Most tumors exhibit a mix of Gleason grades rather than one pure grade. Please explain how authors determined these scores for individual Gleason grades.

Lines 244, 254, 264-266: Figures 4 and 5: What are pre-PCa? Assuming these are pre-cancerous, what is the histopathological classification of these pixels? Were these adjacent to pixels with cancer or isolated in benign regions of the biopsies? Discuss.

In these figures, authors describe Gleason grade 3, grade 4. Again, how did they separate individual grades? Can they produce similar curve for Gleason Grade Group 1 (3+3), 2 (3+4), 3 (4+3), 4 (4+4)? According to the new Gleason Grade Group classifications, risk increase from Grade group 1 to 5. This is of considerable clinical value.

Line 273:  Table 3. It is difficult to interpret sensitivity and specificity results without an ROC curve? What is the area under the ROC curve?  These performance parameters are dependent on selected threshold values to separate benign vs cancer vs Gleason grade3 vs Gleason grade 4. Thus, inclusion of ROC curves will help readers understand performance of the algorithm based on various threshold values.

Author Response

A separate PDF document relating to Reviewer 1 comments has been uploaded. Please let me know if a word file would be more convenient. 

Many thanks for you time, 

Avi Simon 

Reviewer 2 Report

The authors reported a novel modality of prostate cancer, which could use as risk stratification for decisive management and treatment decision. Even if there are many methods to measure the pathological process of prostate cancer, the decreased Zinc modality is adopted for clinical application. The authors introduced this approach with fresh samples, and estimation of gland level of Zinc with compactness for accurate diagnosis. Furthermore, this is essential for accurate diagnosis of prostate cancer with malignancy score.  This study may contribute to the personalized risk stratification and treatment of prostate cancer.

The results and implications of this study will be of interest not only for prostate cancer researchers but more in general to the clinic application. I think the manuscript should be considered for publication, as long as the authors are able to address some specific concerns (see below).

1, The figure 2 is not so clear and seems darker. Could you give more clear figure and maybe also provide the mechanism of the Bruker XRF device?

2, As authors clarify the Zinc approach for accurate diagnosis, it is better to show some samples with different methods for diagnosis by comparing with new approach and other methods such as immunohistochemical analysis, MRI, ultrasonic testing etc. So it is more persuasive for this approach.

3, this manuscript cited 11 reference, and I think that it is better to read more references and compare their results.

4, For the discussion part, I do not fell that authors deep discuss questions of this new approach. There are some list of characteristics and improvements.

5, As the biopsy sample is quite small, results from this sample may not represent the true situation in the prostate. How should doctor accurately diagnose by using this new method?

6, In the discussion part, authors did not cite reference and discuss other people work on the Zinc study on prostate cancer. Could authors update the new progress of this study with more references?

7, The author mentioned that “Accordingly, in case of doubt, the urologist has additional information at his disposal for  getting more decisive and safer decisions at a much lower cost than utilizing expensive and  quite inaccurate genomic, or other risk prognosis methods, that in addition to their inherent inaccuracy can – in the best case – only be as good as the biopsy samples quality. Moreover, genomic tests for lower malignancy score prostates are not as meaningful, while ScoRisk can provide accurate information for low-malignancy and even benign prostates.”

Even authors showed good application in clinic for prostate cancer, the genomic testing is still important in prostate cancer. The genetic of prostate is most important driver of prostate cancer. Expensive and inaccurate of genetic depend the technology development and how many genetic or marker to measure. I think that the authors should cite some reference for “The genomic test are not as meaningful”. Because there are many study showing genetic importance in low-malignancy and even benign prostates.

Author Response

A separate PDF document with our notes for Reviewer 2 has been uploaded. Please let me know if a Word file would be more convenient.

Many thanks for you time,

Avi Simon

Round 2

Reviewer 2 Report

The revised manuscript solved my questions, and I think that it should be accepted.